# Alteration in the Expression of Genes Involved in Cerebral Glucose Metabolism as a Process of Adaptation to Stressful Conditions

**DOI:** 10.3390/brainsci12040498

**Published:** 2022-04-13

**Authors:** Mariola Herbet, Iwona Piątkowska-Chmiel, Monika Motylska, Monika Gawrońska-Grzywacz, Barbara Nieradko-Iwanicka, Jarosław Dudka

**Affiliations:** 1Department of Toxicology, Faculty of Pharmacy, Medical University of Lublin, Jaczewskiego 8b Street, 20-090 Lublin, Poland; iwona.piatkowska-chmiel@umlub.pl (I.P.-C.); m_motylska@onet.pl (M.M.); monika.grzywacz@umlub.pl (M.G.-G.); jaroslaw.dudka@umlub.pl (J.D.); 2Department of Hygiene and Epidemiology, Faculty of Medical Sciences, Medical University of Lublin, Chodźki 7 Street, 20-093 Lublin, Poland; barbaranieradkoiwanicka@umlub.pl

**Keywords:** chronic stress, depression, cerebral glucose metabolism, gene expression

## Abstract

Exposure to chronic stress leads to disturbances in glucose metabolism in the brain, and changes in the functioning of neurons coexisting with the development of depression. The detailed molecular mechanism and cerebral gluconeogenesis during depression are not conclusively established. The aim of the research was to assess the expression of selected genes involved in cerebral glucose metabolism of mice in the validated animal paradigm of chronic stress. To confirm the induction of depression-like disorders, we performed three behavioral tests: sucrose preference test (SPT), forced swim test (FST), and tail suspension test (TST). In order to study the cerebral glucose metabolism of the brain, mRNA levels of the following genes were determined in the prefrontal cortex of mice: *Slc2a3*, *Gapdh*, *Ldha*, *Ldhb*, and *Pkfb3*. It has been shown that exogenous, chronic administration of corticosterone developed a model of depression in behavioral tests. There were statistically significant changes in the mRNA level of the *Slc2a3*, *Ldha*, *Gapdh*, and *Ldhb* genes. The obtained results suggest changes in cerebral glucose metabolism as a process of adaptation to stressful conditions, and may provide the basis for introducing new therapeutic strategies for chronic stress-related depression.

## 1. Introduction

Exposure to chronic stress in civilized countries is constantly increasing. Moderate acute stress may be beneficial to the body by, i.e., increasing hippocampal cell proliferation and neurogenesis [1,2]. On the other hand, long-term stress leads to inhibition of cell proliferation and reduction of neurogenesis in the central nervous system [2]. Chronic stress may cause immunological and hormonal effects, neuroplastic changes resulting in neurogenesis, and impairment of neurotransmission, as well as, through disturbances in brain energy processes that impair synaptic plasticity, lead to numerous functional changes in the brain [3,4,5].

Cells exposed to chronic stress disrupt the glucose supply and metabolism pathways, which can then lead to brain weakness through various biological changes. The brain is highly dependent on glucose for its primary energy source, and tight regulation of glucose metabolism is critical to the physiology of the central nervous system (CNS) [6]. High energy requirements are needed to maintain healthy brain activity. The oxygen consumption in the brain reaches 20% of the oxygen consumption of the whole organism, even though the brain accounts for only 2% of the mass of the entire human body [7,8] These intensive requirements are covered via the adenosine-5′-triphosphate (ATP) production; ATP is the basis for the maintenance of neurons and non-neuronal cells. Glucose is used to provide precursors for the synthesis of neurotransmitters, and also participates in important regulatory functions, including maintaining a membrane concentration gradient, promoting neuroplasticity, managing oxidative stress, learning, and memory [6]. Glucose metabolism is closely linked to the pathways of cell death by enzymes that are involved in glucose metabolism. The brain’s dependence on glucose as the most important fuel is mainly due to the blood–brain barrier (BBB) and its selective permeability to glucose. The penetration of neuroactive substances (e.g., glutamate, aspartate, glycine, D-serine) into the brain is strongly restricted by the BBB; therefore, they must be synthesized from glucose in the brain. It is worth emphasizing that glucose as an energy source is the most important for the brain, but, for example, during strenuous physical activity or prolonged starvation, ketone bodies replace glucose as the main source of energy for the brain. In addition, systemic lactate is taken up and oxidized by the brain, and may also be an important substrate for the brain [9,10].

What is critical in the stress response is the involvement of the hypothalamus, which induces activation of the sympathetic and parasympathetic nervous systems. Released norepinephrine affects glucose metabolism, increasing astrocyte metabolism, lactate production, and nerve glycolysis, and may lead to the glucose transporters activation [11,12]. In response to stress, the activity of the hypothalamic–pituitary adrenal (HPA) axis plays a major role. Stressful stimuli cause corticotropin-releasing hormone (CRH) to be excreted from the hypothalamus, which then reaches the pituitary gland, releasing adrenocorticotropic hormone (ACTH). The adrenal glands are, in turn, stimulated to excrete glucocorticosteroids into the bloodstream (cortisol in humans, corticosterone in rodents). Glucocorticosteroids, on the other hand, can directly influence the energy metabolism of the brain, and can modulate various brain processes [11,13]. Moreover, exposure to chronic stress stimulates overproduction of reactive oxygen species (ROS) in the cerebral cortex, generates oxidative stress, disorders of the mitochondrial membrane, DNA damage, and further, leads to disturbances in brain glucose metabolism and functioning of neurons [14].

Glucose metabolism is closely related to the physiology and function of the brain. Thus, disruption of the brain’s glucose delivery, as the most important fuel, and metabolism pathways may be associated with a variety of pathological brain disorders, including, but not limited to, oxidative stress, neurodegeneration, apoptosis, autophagy, and cognitive dysfunction, and may also lead to depression [15,16]. Research indicates that metabolic disturbances in the brain that lead to dysfunction of neurotransmission, behavior, and cognition play an important role in the pathogenesis of depression [3]. Glucose metabolism is associated with the regulation of cell death, suggesting co-regulation of metabolic and apoptotic pathways [17]. Moreover, neurons are one type of cell for which energy production depends almost exclusively on glucose metabolism [18]. Current research confirms that metabolic disorders in the brain that lead to neurotransmission, behavior, and cognition dysfunction play an important role in the pathogenesis of depression [3]. However, the detailed molecular mechanism and cerebral gluconeogenesis during depression are not conclusively established. Previously, it has been emphasized that the higher brain glucose levels have a memory-enhancing effect [19], whereas the recent studies show that brain hyperglycemia can impair cognitive processes [20]. Thus, the exact neurobiological mechanisms underlying depressive disorders due to abnormalities in brain energy metabolism are still investigated.

Despite much research and significant advances in medical science, the complex regulation of gene expression involved in the regulation of glucose metabolism in the brain remains unclear. An in-depth understanding of these mechanisms could contribute to better insight into the pathophysiology of a wide variety of brain disorders, and could lead to the development of novel treatment strategies. Therefore, our research was aimed to assess the expression of different genes involved in cerebral glucose metabolism of mice under chronic stress conditions. The validated animal paradigm of chronic stress was used to study clinical depression. To confirm the induction of depression-like disorders, we performed three behavioral tests: sucrose preference test (SPT), forced swim test (FST), and tail suspension test (TST). In order to study the cerebral glucose metabolism, mRNA levels of the following genes were determined in the prefrontal cortex of mice: *Slc2a3*, *Gapdh*, *Ldha*, *Ldhb*, and *Pkfb3*. These genes significantly contribute to normal glucose metabolism in the brain, and are involved in many cellular processes, such as the regulation of gene expression in response to chronic stress.

## 2. Materials and Methods

### 2.1. Animals

The experiment was performed on adult, male, 8-week-old Albino Swiss mice, *Mus musculus*, obtained from a licensed breeder of the Center for Experimental Medicine in Lublin, Poland (077—EMC number in Lublin in the Register of Breeders kept by the Minister of Science and Higher Education (Poland)). Following the procedures, the mice were housed in standard conditions in accordance with the Regulation of the Minister of Agriculture and Rural Development of 14 December 2016 on the minimum requirements to be met by the center, and the minimum requirements for the care of animals kept in the center (Journal of Laws, item 2139). Throughout the experiment, the animals were kept in a room with automatically controlled temperature (22 ± 2 °C), relative humidity (50–55%), 15 air changes per hour, and 12/12 h light/dark cycle. Two groups of sixteen mice in total were housed in polycarbonate cages, with unlimited access to water and food. The procedures adopted in all experiments of this study involving the animals and their care were approved by the Local Ethical Committee, and performed in accordance with the applicable European standards for experimental research on animal models (Act of 15 January 2015 on the protection of animals used for scientific or educational purposes; Directive 2010/63/EU of the European Parliament and of the Council of 22 September 2010 on the protection of animals used for scientific purposes). All activities were carried out by qualified personnel. The animals were under the constant care of a veterinarian; every effort was made to minimize the suffering.

### 2.2. Experimental Project

The projekt of the experiment is presented in Figure 1. Mice were divided into 2 groups with 8 mice in each. The animals were assigned to groups randomly. The total number of animals (16) was estimated in accordance with the requirements of statistical analysis, the three Rs (3Rs) and the ARRIVE guidelines (Animal Research: Reporting of In Vivo Experiments). The first group was a research control group administrated with only 0.9% NaCl and 2% Tween 80 (polyoxyethylene glycol sorbitan monooleate). The second group of animals (the “stressed group”) was provided with a daily dose of 0.9% NaCl and 40 mg/kg of corticosterone (*s.c.*) for 21 days. Corticosterone was administered at a fixed volume of 10 mL/kg of the body weight of the mice. Each mouse was weighed to calculate the drug’s dose that was applied. Directly, prior to administration, the corticosterone solution was prepared. The dose of corticosterone 40 mg/kg was diluted in 1 mL/kg of 2% Tween 80 with 0.9% NaCl. The multiple injections of corticosterone lead to excessive activity of the HPA axis, and cause behavioral, as well as neurochemical, changes in the brain of animals adequate to the symptoms and neurochemical changes co-existing with clinical depression in humans [21]. This animal paradigm is a useful mouse model, in which the role of stress can be further explored [22].

### 2.3. Animal Behavioural Tests

#### 2.3.1. Sucrose Preference Test (SPT)

The sucrose preference test was performed 24 h after the last injection of corticosterone, according to Mao et al., but with small modifications [23]. Seventy-two hours before the aforementioned test, the experimental animals adapted to 2% sucrose solution (*w*/*v*). First, two bottles with this solution were placed in each cage for 24 h, and next, one of them was changed with water for one day. In order to avoid possible side effects when drinking, the bottles were repositioned after 12 h. After adaptation, the mice were deprived of water and food for the next day. The sucrose preference test was performed at 9:00 am. The mice had free access to two bottles containing 100 mL of water and 100 mL of a 2% sucrose solution, respectively, and were kept in separate cages. After 24 h, the volumes (in mL) of water, as well as 2% solution of sucrose, consumed by the animals were recorded. The preference of sucrose was calculated according to the following formula:(sucrose uptake (mL))/(water uptake (mL) + sucrose intake (mL)) × 100%

#### 2.3.2. Forced Swim Test (FST)

The stress ability to depressive behaviors was confirmed by the Forced Swim Test, called the abandonment test, according to Porsolt et al. (1977) [24]. Mice from the control and stress groups were immersed individually in a cylindrical glass vessel (height, 25 cm; diameter, 10 cm) filled with water to a height of 10–11 cm (water temperature: 23–25 °C) for 6 min. The immobility time between 2 and 6 min after immersion was measured using a totalizer. Then, the animals were taken out of the utensil, dried with lignin, and transferred to a cage (dry litter) over which an infrared lamp was placed. The mouse was evaluated as stationary when it ceased fighting and hovered in the water, making only the movements necessary to keep its head above the water level. The immobility time was assessed in real time by two blind observers. Throughout the test, the mice were kept in constant conditions, i.e., temperature, noise, and lighting, and free (except for 6 min of swimming) access to food and water [25]. When the observed animal is fighting, it is interpreted as “coping with stress” behavior, whereas stillness is considered passive behavior that mirrors depressive symptoms [26]. The results of the FST test are shown as the mean of the immobility time of the mice in seconds ± standard error of the mean for the control group and the corticosterone-administered group.

#### 2.3.3. Tail Suspension Test (TST)

The procedure was performed according to Ster et al. (1985) [27]. Each mouse was suspended by its tail on a vertical pole in a wooden box (30 × 30 cm). The animals were attached for 6 min with adhesive tape, fixed 2 cm from the end of the tail. The test lasted 6 min; the total immobility time of the animals was recorded during the last 4 min. The mouse was considered immobile if it hovered motionless, stopped moving its body and limbs, and made only the movements necessary for breathing [28]. The immobility time (in seconds) was assessed by two blind observers (real time); results are expressed as arithmetic mean ± standard error of the mean (SEM) for each experimental group.

### 2.4. Collection of Tissues

Immediately after the behavioral tests, the mice were decapitated. The above method is an acceptable method of rodent euthanasia and complies with European and Polish regulations. Entire brains were carefully removed and rinsed in ice-cold saline to remove blood. The prefrontal cortex was then separated, divided into parts, and frozen at −80 °C.

### 2.5. The Quantitative Real-Time PCR Analysis (qRT-PCR)

#### 2.5.1. RNA Isolation

Total RNA was isolated from 30 mg of the prefrontal cortex of mice by the method of Chomczyński and Sacchi [29] using the TRIzol reagent (Invitrogen, Carlsbad, CA, USA) according to the manufacturer’s instructions. RNA concentration and purity were measured spectrophotometrically using a MaestroNano NanoDrop spectrophotometer (Maestrogen, Hsinchu, Taiwan). Only high purity RNA was used for further studies (A260/280 ratio ranged between 1.8 and 2.0).

#### 2.5.2. cDNA Synthesis

cDNA synthesis was performed using a cDNA reverse transcription kit (Applied Biosystems, Foster City, CA, USA) according to the manufacturer’s instructions. Reaction conditions: 25 °C for 10 min, 37 °C for 120 min, and then 85 °C for 5 min to complete the process. The obtained cDNA was stored at −20 °C.

#### 2.5.3. Real-Time PCR

The relative expression of the following genes was measured: *Slc2a3*, *Gapdh*, *Ldha*, *Ldhb*, and *Pkfb3* by real-time PCR, ΔΔCt, using *Hprt* and *Tbp* as endogenous controls (Table 1). The reaction was performed in successive triplicates using a 7500 Fast Real-Time PCR System (Applied Biosystems, Foster City, CA, USA) and Fast Probe qPCR Master Mix (2×) plus ROX solution (EURx, Gdańsk, Poland). The reaction mixture contained 10 µL of Fast Probe qPCR Master Mix (2×), 9 µL of RNase-free water, ROX solution (50 nM), and 0.5 µM of the gene-specific TaqMan probe (Applied Biosystems, Foster City, CA, USA). Data quality screening was performed based on amplification, Tm, and Ct values to remove any outliers before calculating ΔΔCt, and determining the fold change in mRNA levels. Data are presented as RQ value (RQ = 2 − ΔΔCt).

### 2.6. Statistical Analysis

Data was analyzed by STATISTICA v.10 application (StaftSoft, Cracow, Poland). The results are presented as mean ± SEM. The normality of the distribution was analyzed using the Shapiro–Wilk test. Comparisons between groups were done using a Student’s *t*-test or Mann–Whitney U test (depending on normality of distribution), and *p* < 0.05 was considered to indicate a statistically significant difference.

## 3. Results

### 3.1. Body Weight of Mice

The results of the obtained body mass of rodents are presented in the Table 2. Our study showed that the 21-day administration of corticosterone resulted in a statistically significantly lower weight gain on day 7, 14, and 21 of the measurement when compared to the control group.

### 3.2. Consumption of Sucrose

Figure 1a shows the results of the sucrose consumption test expressed as a percentage of the consumption of 2% sucrose solution to the sum of the consumption of water and 2% sucrose solution. Our study revealed that repeated administration of corticosterone led to a significant decrease of the percentage of sucrose consumption by the animals (*t* = 3.942, *p* < 0.01), when compared to control.

### 3.3. Immobility Time in the FST and TST

The chronic administration of corticosterone to mice resulted in a significant increase in immobility in the FST (*t* = 3.818, *p* < 0.01, Figure 1b), as well as in the TST (*t* = 2.632, *p* < 0.05, Figure 1c), in comparison with control.

### 3.4. Gene Expression Studies

In the group of animals that received corticosterone 40 mg/kg for 21 days, a statistically significant increase in the expression of the Slc2a3 (*t* = 2.331, *p* < 0.05, Figure 2a) and Ldha (*t* = 4.706, *p* < 0.001, Figure 2c) genes was noted in comparison with the control group. In turn, in the animals chronically injected with corticosterone, a statistically significant reduction in the mRNA expression of the Gapdh (*t* = 2.616, *p* < 0.05, Figure 2b) and Ldhb (*t* = 2.979, *p* < 0.01, Figure 2d) genes was noted as compared to the animals from the vehicle group. In the group of mice exposed to chronic stress, no statistically significant changes were noted in the expression of the Pfkfb3 gene (*p* > 0.05, Figure 2e).

## 4. Discussion

Due to high energy requirements of the active neurons, it is necessary to ensure an adequate supply of oxygen and glucose. Chronic stress disrupts homeostasis, making the brain unable to cope with glucose catabolism. The reason lies in the disruption of both glucose supply to neurons, and glucose metabolism present in neurons. However, there is a limited number of studies concerning these issues. It is thus important to assess the expression of metabolic genes involved in the cerebral glycolytic pathway, as it may enable a new perspective on the pathogenesis of depression, and thus, the selection of the appropriate path of treatment. In the present work, the expression of selected genes associated with the brain energy metabolism in mice exposed to chronic stress was evaluated. Three behavioral tests were performed to confirm the adopted model of depression, and included established paradigms: SPT, an indicator of anhedonia-like behavioral change; and also, FST and TST—the behavioral despair tests.

Our findings showed that exogenous chronic corticosterone administration resulted in significantly slower weight gain in mice as compared to the control group. Moreover, in behavioral tests, it was shown that corticosterone administered chronically develops a model of depression. Mice receiving corticosterone consumed statistically significantly less sucrose solution as compared to the control group. These results are compatible with the literature data, showing that the repeated corticosterone injections result in decreased responsiveness to rewards, as reflected by the decrease in the percentage of sucrose consumption of mice [30]. Furthermore, the animals subjected to chronic corticosterone injections exhibited significant behavioral despair, as shown by the significantly increased immobility time in both FST and TST. The decreased weight gain of the animals, as well as the observed changes in their behavior, may confirm that chronic repeated administration of corticosterone causes stress reactions in animals. These changes also confirm the depression model adopted in the experiment, and may constitute the basis for the molecular analysis of the organism’s adaptation strategy to environmental variability.

The obtained results show statistically significant changes, increasing the level of mRNA expression of *Slc2a3* and *Ldha* genes in the stressed group compared to the control group. A statistically noticeable decrease in expression was reported for the *Gapdh* and *Ldhb* genes, contrary to lack of changes in the *Pkfb3* gene.

Solute carrier family 2 member 3 (SLC2A3) is a gene which encodes SLC2A3 protein and then GLUT3. This is a group of proteins that are particularly significant for brain metabolism. As mentioned in the introduction, proper functioning of neurons depends on brain metabolism, including the synthesis of neurotransmitters, i.e., glutamate and gamma aminobutyric acid (GABA). The size of those molecules is too large to pass through the blood–brain barrier (BBB); therefore, they must be synthesized in the brain from glucose. The GLUT3 allows glucose cross by facilitative diffusion through cell membranes, such as BBB, which helps glucose to get into the nerves. GLUT3 is mainly found in neuron axons and dendrites. A constant supply of GLUT3 for brain work must be ensured because, compared to other glucose transporters, this one is the most numerous in the cerebral cortex, and has the greatest affinity for glucose. It is important because brain activity needs high energy delivery. Brain glucose uptake is thought to be independent of insulin [31,32,33,34]. SLC2A3 significantly contributes to normal glucose metabolism in the brain [33]. According to studies, the effect of chronic stress on GLUT3 expression is visible in the hippocampus, amygdala, and hypothalamus—without changes in the prefrontal cortex [35]. Previous studies have shown that hunger and associated periodic hypoglycemia, as well as brain pathologies such as hypoxia, induce an increase in GLUT3 protein levels in the mouse brain. Therefore, even with mild hypoglycemia, the substrates necessary for the production of neurotransmitters and the function of neurons are always delivered to the brain [36]. Accordingly, chronic, stress-induced increases in ROS can damage various macromolecules in the cell, including those that make up the electron transport system. This results in damage to the mitochondria, and the formation of energy deficits—one of the adaptation pathways will then be the stimulation of glucose uptake. Moreover, there is a correlation between glucose uptake and ROS production and uptake; thus, maintaining redox status depends on glucose transporters [32]. The overexpression of GLUT3 in our study probably results from the need to restore cellular energy homeostasis in the brain of mice exposed to chronic stress. The oxidation of more glucose increases the chances of compensating for the energy deficit, which results in supporting the energy metabolism in the cerebral cortex. The glucose transport capacity exceeds demand in a wide range, and the very high GLUT3 transport rate ensures that the neurons receive sufficient glucose stores at different glucose levels and different activity states [37].

A reduction in the *Gapdh* gene expression level was observed for the group of mice under chronic stress compared to the control group. Glyceraldehyde-3-phosphate dehydrogenase (GAPDH/G3PDH) gene encodes a protein-including family of enzymes that take part in carbohydrate metabolism. This enzyme is from the class of oxidoreductases, and takes part in the sixth step of glycolysis. GAPDH converts 3-phosphoglycerol aldehyde (G3P) into 1,3-diphosphoglycerate (BPG) by adding phosphate and removing hydrogen in the attendance of nicotinamide adenine dinucleotide (NAD) [38]. Moreover, GAPDH is called a “moonlighting protein”, directly involved in many cellular processes, such as regulation of gene expression, response to oxidative stress, DNA repair, and apoptosis. The enzyme participates in many processes due to its movement between the cytosol and the nucleus in the cell and post-translational modifications, including S-thiolation, S-nitrosylation, and phosphorylation [39]. Recent studies have also shown that the GAPDH enzyme is located on axonal vesicles, and is necessary and sufficient for providing energy in the fast axonal transport (FAT), depending on glycolytic ATP [40]. According to many researchers, oxidative stress triggering reactive oxygen and nitrogen forms is a factor that causes numerous modifications and damage to the enzyme 3-phosphoglyceroldehyde dehydrogenase. The oxidative modification, i.e., S-thiolation or S-nitrosylation, refers to the cysteine rest 152 located in the active center of the enzyme, which translates into a decrease in the activity of the enzyme [41]. This phenomenon occurs in neurodegenerative diseases when GAPDH is overexpressed with a simultaneous decrease in glycolytic activity of the enzyme [42]. GAPDH participates in the last stage of glycolysis, and is responsible for the conversion of G3P to BPG using NAD as a cofactor. Chronic stress states increase the amount of ROS in the brain, which forces the activation of appropriate defense pathways. The significant reduction in the mRNA expression of the *Gapdh* observed in our study indicates a decrease in the activity of glycolytic enzymes, and is probably the result of the redirection of carbohydrate catabolism in cells from glycolysis to the pentose phosphate pathway during nicotinamide adenine dinucleotide phosphate (NADPH) generation. The final evidence for the role of NADPH was provided by the value of the NADPH/NADP^+^ ratio, which is important for maintaining the redox balance and antioxidant activity. Unfortunately, chronic inhibition of glycolysis mechanisms and antioxidant activity may eventually be overcome by the increasing amount of ROS in the brain. As a result, cell death may occur, indicative of the apoptotic function of GAPDH. Numerous studies have been carried out, among which, it has been proven that GAPDH accumulates in mitochondria, increasing their membrane permeability, which is a factor inducing apoptosis. According to other studies, a decrease in the level of mRNA expression for the GAPDH gene may also indicate mitochondrial Ca^2+^ retention. This reduces the amount of released neurotransmitter, and creates conditions for cell apoptosis [43,44]. Taking into account both the function and localization of GAPDH in the cell and the obtained results, it can be concluded that GAPDH expression is altered in the states of chronic stress, and it can also be assumed that these mechanisms play a role in depression. Studies have shown that GAPDH inhibits cell death under certain conditions [45]. Other studies suggest that GAPDH mediates neuronal apoptosis, which may be related to DNA damage [46].

A statistically significant increase in the mRNA expression of *Ldha* gene was observed among chronically stressed mice. LDHA (lactate dehydrogenase) gene encodes isoform A of LDH enzyme, one of the five different forms of this enzyme. That protein is found in the whole organism, and is significant for performing a chemical reaction role in energy metabolism. It catalyzes the last step of the glycolysis–anaerobic metabolic pathway [47]. The A subunit is primarily found in skeletal muscle. For high-intensity physical activity, muscles need large amounts of energy, and the body’s oxygen intake may not be enough for the amount of energy required. To this end, glycogen is broken down to glucose, and during the final stage of glycogen breakdown, the LDHA converts a pyruvate molecule into a molecule called lactate, which can be used as an energy source. It happens with concomitant inter-conversion of NADH to NAD^+^ [48]. Lactate synthesis in the brain takes place in astrocytes. Astrocytes release lactate faster than neurons, which is why it can be consumed as energy fuel for neurons. The neuronal conversion of lactate to pyruvate does not require ATP. Neurons have been shown to contain mainly LDH-1, which acts “towards” pyruvate, whereas astrocytes express LDH-5 and act “towards” lactate [49]. According to the Astrocyte-Neuron Lactate Shuttle Hypothesis in resting conditions, neurons utilize glucose, whereas during increased energy demand, they preferentially choose lactate. This is important for the production of functional neurological signals and cerebrovascular coupling [50,51]. Recent studies found that lower levels of LDH among patients suffering from depression compared to controls was statistically significant, but further studies are still needed [48]. LDHA is important from the point of view of energy metabolism in the brain. LDHA regenerates NAD^+^ with NADH to maintain glycolysis, producing lactate as a by-product from pyruvate. LDHA is present in astrocytes, but not in neurons; thus, an increase in LDHA expression activates LDH enzymes in astrocytes, leading to an increase in lactate levels in the brain. Lactate is easily available because astrocytes have the ability to release it quickly. The increase in the *Ldha* gene expression in our study may indicate that under increased energy demand conditions, resulting from, e.g., chronic stress, neurons choose lactate consumption, whereas astrocytes metabolize more pyruvate, and thus, provide neuronal substrates. This may indicate a “glucose sparing” physiological state in which the use of additional oxidative fuel helps to maintain glucose availability to the glycolytic and pentose phosphate transfusion pathways that provide critical brain functions under conditions of chronic stress. Under chronic stress conditions, cellular respiration in the brain’s mitochondria is insufficient to meet energy needs due to mitochondrial dysfunction. Thus, energy metabolism is redirected to the anaerobic pathway, which may enable normal neuronal transmission in the brain [48,50,51].

In our study, the expression of the *Ldhb* gene level was, in turn, statistically significantly decreased in the cortex of mice receiving corticosterone compared to the control group. The lactate dehydrogenase B (LDHB) gene encodes the B subunit of the LDH enzyme. LDHB is the enzyme that catalyzes the interconversion of pyruvate and lactate. At the same time, the mutual conversion of NADH to NAD^+^ takes place in the process of glycolysis. The LDHB subunit, similarly to the aforementioned LDHA, generates five tetrameric isoenzymes [52]. LDHB is responsible for the conversion of lactate to pyruvate, with the reciprocal use of NAD^+^ necessary for the regeneration of NADH. Chronic stress inducing an increase in ROS reduces oxygen supply and mitochondrial function. The importance of reduced *Ldhb* mRNA levels in chronic stress may be because of the optimization of lactate uptake for ATP production rather than conversion to pyruvate. This could be due to the overproduction of NADH by a defense mechanism due to the current oxidative stress. The results obtained in our study suggest that LDH is involved in the control of the intracellular redox state. This supports the hypothesis that an increase in the amount of mitochondrial ROS stimulates mitochondrial dysfunction in the brain, which, in turn, results from an increase in the level of lactate in the cerebral cortex under conditions of chronic stress. Lactate is an alternative source of a significant amount of ATP for neurons after the brain has been subjected to prolonged stress, and this may be confirmed by modulation of *Ldhb* gene expression. Similar results were obtained using the rat chronic stress model [32].

PFKFB3 gene (6-phosphofructo-2-kinase/fructose-2, 6-biphosphatase) has been identified as a glycolysis control gene. The encoded protein belongs to the group of bifunctional proteins that play a role in the modulation of the regulatory molecule, not only through the synthesis, but also the degradation of fructose-2,6-bisphosphate (F2, 6BP), which is necessary for the control of glycolysis in eukaryotes. The biosynthesis of F2BP2 is dependent on the activity of the PFKFB3 protein, which acts as a potent allosteric activator for phospho-6-kinase-1. On the other hand, the activity of 6-phospho-fructo-1-kinase has a decisive influence on the course of glycolysis [53,54]. In neurons, glucose metabolism mainly takes place via the pentose phosphate pathway, as it enables the regeneration of NADPH (H) and the support of the neuronal redox state. Excitotoxicity is associated with excessive N-methyl-D-aspartate receptor (NMDAR) excitation, which stabilizes the PFKFB3 protein in neurons, and redirects the glucose pathway from the pentose–phosphate pathway (PPP) to glycolysis. Consequently, this results in low availability of NADPH (H), which is used for the correct regeneration of GSH. This situation generates oxidative stress, and leads to the death of neurons. Importantly, silencing of the PFKFB3 protein in neurons prevents the intensification of ROS production and apoptosis induced by excitotoxic factors [54]. The coding proteins play a regulatory role, which is believed to be important for MDD under stress-related conditions. Studies have shown that proteasomal degradation of the PFKFB3 protein by the ligase, anaphase/cyclosome promoting complex (APC/C-Cdh1) redirects glucose catabolism from glycolysis to PPP [55]. This ensures an increase in the production of NAPDH, and enables the maintenance of the redox balance in the intracellular environment under conditions of chronic stress and nutrient deficiency. Therefore, by actively lowering glycolysis, the neurons in the cerebral cortex use glucose to maintain redox balance, limiting its use for bioenergy purposes. Studies have also shown that the NMDAR-stimulation-induced shift of PPP to glycolysis occurred under oxidative stress, as evidenced by an increase in redox oxidized glutathione, an increase in mitochondrial ROS, and neuronal apoptosis [54,56]. Ultimately, it can induce oxidative stress, and then lead to neuronal apoptosis due to insufficient PPP—processes characteristic for the pathogenesis of depression. In our study, no statistically significant changes in the mRNA expression level for the *Pfkfb3* gene were observed. This may be a consequence of the activation of NMDA subtypes of glutamate receptors, which may prevent degradation of the PFKFB3 protein in cortical neurons. The lack of changes in gene expression may also suggest that glycolysis in the brain of mice exposed to chronic stress is sufficiently controlled at the recorded level.

## 5. Conclusions

The overexpression of *Slc2a3* observed in our study indicates the role of GLUT3 in maintaining cellular energy homeostasis, and supplying more energy to neurons subjected to chronic stress. Changes in the expression of the *Ldha* and *Ldhb* genes may, in turn, indicate the brain’s ability to produce and use lactate to optimize energy use and metabolism under stressful conditions. On the other hand, the decrease in the expression of the *Gapdh* gene may indicate a redirection of energy metabolism from anaerobic glycolysis to the pentose–phosphate pathway. The obtained results suggest changes in the cerebral glucose metabolism, as a process of adaptation to stressful conditions, indicate a significant role of the studied genes in the pathogenesis of stress-related depression, and may constitute the basis for the development of new therapeutic solutions for depression associated with chronic stress.

## Data Availability

Not applicable.

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
