# Peer review of "Alteration in the Expression of Genes Involved in Cerebral Glucose Metabolism as a Process of Adaptation to Stressful Conditions"

_brainsci, 2022, doi:10.3390/brainsci12040498_

Round 1

Reviewer 1 Report

Herbet et al has demonstrated the disruption of energy metabolism in the brain of corticosterone induced mouse model of chronic stress. Brain consumes twenty percent of the total oxygen requirements of the body and produces ATP through the glucose metabolism and oxidative phosphorylation in the mitochondria of the brain. Authors have particularly focused on glucose metabolism, whereas there are amino acid and lipid metabolism also contribute to the energy/ ATP production in the brain. Authors have mentioned that “In order to study the brain energy metabolism, mRNA levels of the following genes were determined in the prefrontal cortex of mice: Slc2a3, Gapdh, Ldha, Ldhb and Pkfb3”. Authors have not elaborated the idea of choosing/ picking up selected gene for targeted expression study, whereas several other genes also involved in the glucose metabolism in the brain. This paper portrays a part of the energy metabolism altered in the mouse model of chronic stress. This paper is not matured enough for the publication.

  1. Authors did not mention the age of the animals at the time of experiment. How did the author standardize the dose of corticosterone for disease model? Duration of the treatment is missing.
  2. Did the author measure APT levels in the brain of the experimental mice?
  3. Did authors find any alteration of HPA axis in the experimental mice? Did authors measure the hormonal levels in the mice?
  4. Not only the glucose metabolism, amino acid and lipid metabolism are also the source of ATP production in the brain. Several studies have shown that amino acid metabolic pathways are altered in the stressed condition of the brain. Therefore, focusing on glucose metabolism only represent a part of altered metabolic pathways. If author want to focus on glucose metabolism during stressed condition, authors need to frame the manuscript accordingly. This form of the manuscript is not conveying the full message about disruption of energy metabolism in the brain of mice during stressed condition.
  5. GAPDH is considered as a housekeeping gene and used for normalizing other gene expression. Here authors showed that in stressed condition mRNA of GAPDH has significantly decreased. How did author normalize the gene expression in the brain tissue?

Author Response

Response to Reviewer 1 Comments

Herbet et al has demonstrated the disruption of energy metabolism in the brain of corticosterone induced mouse model of chronic stress. Brain consumes twenty percent of the total oxygen requirements of the body and produces ATP through the glucose metabolism and oxidative phosphorylation in the mitochondria of the brain. Authors have particularly focused on glucose metabolism, whereas there are amino acid and lipid metabolism also contribute to the energy/ ATP production in the brain. Authors have mentioned that “In order to study the brain energy metabolism, mRNA levels of the following genes were determined in the prefrontal cortex of mice: Slc2a3, Gapdh, Ldha, Ldhb and Pkfb3”. Authors have not elaborated the idea of choosing/ picking up selected gene for targeted expression study, whereas several other genes also involved in the glucose metabolism in the brain. This paper portrays a part of the energy metabolism altered in the mouse model of chronic stress. This paper is not matured enough for the publication.

Point 1: Authors did not mention the age of the animals at the time of experiment. How did the author standardize the dose of corticosterone for disease model? Duration of the treatment is missing.

Response 1: We would like to thank the Reviewer for comments and suggestion, which helped us improve the quality of our manuscript. The revised version of the manuscript included the missing information: we reported the age of the animals (8-week-old), duration of treatment (21 days), and information on how the mice were administered corticosterone. The dose of corticosterone was selected based on the literature and our previous experience. These experiments confirm that the dose and timing of corticosterone administration are sufficient to achieve the goal of neurochemical and behavioural changes in animals adequate to the symptoms coexisting with clinical depression in humans.

Araki R, Tachioka H, Kita A, Fujiwara H, Toume K, Matsumoto K, Yabe T. Kihito prevents corticosterone-induced brain dysfunctions in mice. J Tradit Complement Med. 2021 May 15;11(6):513-519. doi: 10.1016/j.jtcme.2021.05.002. PMID: 34765515; PMCID: PMC8572719.

Weng L, Guo X, Li Y, Yang X, Han Y. Apigenin reverses depression-like behavior induced by chronic corticosterone treatment in mice. Eur J Pharmacol. 2016 Mar 5;774:50-4. doi: 10.1016/j.ejphar.2016.01.015. Epub 2016 Jan 27. PMID: 26826594.

Herbet M, Natorska-Chomicka D, Ostrowska M, Gawrońska-Grzywacz M, Izdebska M, Piątkowska-Chmiel I, Korga A, Wróbel A, Dudka J. Edaravone presents antidepressant-like activity in corticosterone model of depression in mice with possible role of Fkbp5, Comt, Adora1 and Slc6a15 genes. Toxicol Appl Pharmacol. 2019 Oct 1;380:114689. doi: 10.1016/j.taap.2019.114689. Epub 2019 Jul 22. PMID: 31344373.

Point 2: Did the author measure APT levels in the brain of the experimental mice?

Response 2: In our study, we didn't evaluate the level of ATP in the brain of mice. Due to the limited availability of mice brain sample, the ATP could not be measured. In next experiment we have planned brain ATP level measure, given the undeniable role of ATP in the CNS, and also   the relationship of depression with level of ATP production.

Gardner A, Johansson A, Wibom R, Nennesmo I, von Döbeln U, Hagenfeldt L, Hällström T. Alterations of mitochondrial function and correlations with personality traits in selected major depressive disorder patients. J Affect Disord. 2003 Sep;76(1-3):55-68. doi: 10.1016/s0165-0327(02)00067-8. PMID: 12943934.

Rezin GT, Cardoso MR, Gonçalves CL, Scaini G, Fraga DB, Riegel RE, Comim CM, Quevedo J, Streck EL. Inhibition of mitochondrial respiratory chain in brain of rats subjected to an experimental model of depression. Neurochem Int. 2008 Dec;53(6-8):395-400. doi: 10.1016/j.neuint.2008.09.012. Epub 2008 Sep 27. PMID: 18940214.

Point 3: Did authors find any alteration of HPA axis in the experimental mice? Did authors measure the hormonal levels in the mice?

Response 3: Since we used an animal model in which mice were administered corticosterone for 21 days, we did not plan to determine its level after the end of the experiment. The obtained results could not be clearly interpreted. It would be unclear what the corticosterone pool is due to exogenous administration and what is the endogenous hormone pool. Behavioural tests and body weight measurements performed after its completion were the confirmation of the assumed changes in the adopted model (described in the scientific literature). We have adopted the hypothesis that the use of a proven animal model, confirmed by behavioural research, leads to changes in the level of hormones in the body, which has been the subject of many studies. The aim of our study was to evaluate the expression of selected genes involved in glucose metabolism in the brain during chronic stress. Such research is limited and the scientific literature lacks detailed data on this subject. We agree with the Reviewer that the determination of the levels of hormones in animals would be an additional, valuable confirmation of our assumptions, however, they were not made. It is currently not possible to perform additional determinations; we will consider this consideration in the planning of future experiments. However, in our opinion, the results obtained in the study are valuable, provide new information and should be published.

Point 4: Not only the glucose metabolism, amino acid and lipid metabolism are also the source of ATP production in the brain. Several studies have shown that amino acid metabolic pathways are altered in the stressed condition of the brain. Therefore, focusing on glucose metabolism only represent a part of altered metabolic pathways. If author want to focus on glucose metabolism during stressed condition, authors need to frame the manuscript accordingly. This form of the manuscript is not conveying the full message about disruption of energy metabolism in the brain of mice during stressed condition.

Response 4: After considering the comments of the Reviewers, we decided to re-edit the title of the work and the introduction. Our research focused on glucose-dependent brain energy metabolism. However, this was not emphasized in the original version of the manuscript. Therefore, in this new version, we focused on cerebral glucose metabolism in mice during stressed condition; we also mentioned alternative energy sources for the brain.

Point 5: GAPDH is considered as a housekeeping gene and used for normalizing other gene expression. Here authors showed that in stressed condition mRNA of GAPDH has significantly decreased. How did author normalize the gene expression in the brain tissue?

Response 5: We thank the Reviewer for raising such an extremely important issue. The reviewer is right, Gapdh is considered as a housekeeping gene and routinely is being used for normalizing the gene expression. However, given that Gapdh protein is a catalytic enzyme involved in glycolysis and is therefore important for cell metabolism, it should be considered whether Gapdh is a housekeeping gene.

A feature of a good reference gene is its constant expression independent of the conditions of the conducted experiment. As the Reviewer noted, in the conditions of our experience Gapdh expression changed. Therefore, its activity has been shown to be affected. In our opinion, the reference gene should be selected each time for a given experiment. The preliminary research (not published data), a number of reference gene candidates were examined in the prefrontal cortex of mice used in this experiment (from both, control and study groups).

The results of gene expression were normalized with the use of reference genes with the lowest standard deviation from Ct in the tested samples (Hprt1 and Tbp, as mentioned in methods 2.5.3. section), in accordance with the MIQE guidelines (minimum information for publication of quantitative real-time PCR experiments) [Bustin 2009].

Bustin SA, Benes V, Garson JA, Hellemans J, Huggett J, Kubista M, Mueller R, Nolan T, Pfaffl MW, Shipley GL, Vandesompele J, Wittwer CT. The MIQE guidelines: minimum information for publication of quantitative real-time PCR experiments. Clin Chem. 2009;55:611-22.

Reviewer 2 Report

This study aims to assess the expression of selected genes involved in brain energy metabolism of mice in the validated animal paradigm of chronic stress. The study is interesting but it presents some faults in the design of the experiment and data analysis.

1.-Why has a sample size of 16 mice been chosen?. It is not established in the materials and methods section if the sample size has been determined according to the minimum significant difference to be detected in the experiment or if it has been for ethical reasons or both.

2.- It is also not described how the mice were assigned to the experimental groups, was the allocation randomized? It would be interesting to know whether the average weight of the mice in the two experimental groups and what other characteristics of the rats may affect the stress of such mice. It is very important in a design that the groups are comparable to avoid confounding factors.

3.- Means and standard errors have been used to describe the values of the variables measured in each experimental group and a Student's t-test has been carried out to compare the means of the groups. I think it is necessary to describe the distribution of the data. If the distribution of the data in the different variables is asymmetric, the t-test would not be the most powerful test to detect differences. Just as the mean would not be the most representative measure to describe the data. Even if there is an outlier it may affect the mean due to the small sample size. It would be interesting to visualize the distribution of the data by box-plot in both experimental groups.

4.- When the authors show the results of the statistical tests they indicate an F-value, t-value, and p-value. I do not know what the F-value refers to. If this value refers to the test of equality of population variances, another p-value would be necessary for this test. It would be important to specify the meaning of the F-value.

5.- The abbreviations should be revised because the rule that they should be indicated when they appear for the first time, even if they are known, is not complied with. For example, on page 2, line 57, the abbreviation ROS appears but it is not indicated what it refers to. In the section on statistical analysis, it is not indicated that it is SEM. I recognize that this abbreviation is well known but I think that the full term should always be indicated to avoid confunding factors.

Author Response

Response to Reviewer 2 Comments

This study aims to assess the expression of selected genes involved in brain energy metabolism of mice in the validated animal paradigm of chronic stress. The study is interesting but it presents some faults in the design of the experiment and data analysis.

Point 1: Why has a sample size of 16 mice been chosen?. It is not established in the materials and methods section if the sample size has been determined according to the minimum significant difference to be detected in the experiment or if it has been for ethical reasons or both.

Response 1: We would like to thank the Reviewer for comments and suggestion, which helped us improve the quality of our manuscript. The total number of animals was estimated in accordance with the requirements of statistical analyses, the Three Rs (3Rs) and the ARRIVE guidelines (Animal Research: Reporting of In Vivo Experiments). This sentence was included in the revised manuscript.

Point 2: It is also not described how the mice were assigned to the experimental groups, was the allocation randomized? It would be interesting to know whether the average weight of the mice in the two experimental groups and what other characteristics of the rats may affect the stress of such mice. It is very important in a design that the groups are comparable to avoid confounding factors.

Response 2: The animals were assigned to groups randomly. This sentence also was included in the revised version of manuscript. The mice were weighed each day just before corticosterone administration and the body weights of the mice were recorded every 7 days. We observed that the stressed mice did not gain weight as compared to the control group. We subjected the obtained results to statistical analysis and included them in the form of a table in the manuscript (Table 2).

Point 3: Means and standard errors have been used to describe the values of the variables measured in each experimental group and a Student's t-test has been carried out to compare the means of the groups. I think it is necessary to describe the distribution of the data. If the distribution of the data in the different variables is asymmetric, the t-test would not be the most powerful test to detect differences. Just as the mean would not be the most representative measure to describe the data. Even if there is an outlier it may affect the mean due to the small sample size. It would be interesting to visualize the distribution of the data by box-plot in both experimental groups.

Response 3: The normality of distribution was analysed using the Shapiro-Wilk test. All variables except variable Pkfb3 had normal distribution. Therefore, the use of Student's t-test was justified. In the variable Pfkb3, the non-parametric test which is the Mann-Whitney U test was used. This has been corrected in the manuscript. Due to the single number of outlier variables and the lack of extremely outlier variables, it was decided that the mean and standard errors would be a better measure of the data set. Moreover, mean and median values were similar to each other. We agree with the Reviewer that visualizing the data distribution according to the box plot in both experimental groups could provide better insight into the results of the statistical analysis. Below we present the charts made in line with the Reviewer's suggestion. However, in our opinion, column charts are more readable and accessible to non-specialist readers, especially in the field of statistical methods. Additionally, taking into account the fact, that in our study we did not observe extreme outliers, we decided to leave the column graphs in the manuscript.

Point 4: When the authors show the results of the statistical tests they indicate an F-value, t-value, and p-value. I do not know what the F-value refers to. If this value refers to the test of equality of population variances, another p-value would be necessary for this test. It would be important to specify the meaning of the F-value.

Response 4: The F-value was used unnecessarily in this study. We would like to thank the Reviewer for this remark. We have deleted this data in the manuscript.

Point 5: The abbreviations should be revised because the rule that they should be indicated when they appear for the first time, even if they are known, is not complied with. For example, on page 2, line 57, the abbreviation ROS appears but it is not indicated what it refers to. In the section on statistical analysis, it is not indicated that it is SEM. I recognize that this abbreviation is well known but I think that the full term should always be indicated to avoid confunding factors.

Response 5: In accordance with the Reviewer's remark and the applicable rule, the abbreviations have been revised and supplemented.

Reviewer 3 Report

In the paper by Mariola Herbet et al., titled “Alteration in the expression of genes involved in brain energy metabolism as a process of adaptation to stressful conditions” the authors describe changes in brain energy metabolism as a process of adaptation to stressful conditions and may constitute the basis for the development of new therapeutic solutions for depression associated with chronic stress.

The article is written on an intriguing topic of modern neuroscience and has a scientific interest.

However, several issues need to be addressed.

Major issues:

  1. According to the authors, studying the expression of genes involved in energy brain metabolism, it can be concluded that they are involved in adaptation to stressful conditions.The authors are welcome to present more data on changes in glucose/energy metabolism (expression of enzymes involved in the regulation of energy metabolism, lactate, mitochondria, etc.).
  2. The introduction is very long, but does not specify what the focus will be on.It makes sense to describe in more detail the choice of genes for research and describe what mechanism they plan to study.
  3. In the introduction, it makes sense to supplement information about the current state of the problem of energy metabolism in the brain.
  4. Line 81-93 – clearly highlight the main idea and focus of the study
  5. It is not entirely clear the choice of the stress model.Why did you choose a corticosteron administration, and not other models (early-life stress etc).Reference to article [17] did not provide a description of the model.Please clarify.
  6. 2 Experimental project: I recommend adding a design scheme. The description of the stress model is best placed in the methods section, rather than in the introduction.
  7. Why gene expression was assessed only in the cortex and not in the hypothalamus, pituitary gland, amygdala (fear) or other regions?
  8. Did you measure basal levels of corticosterone and glucose and after modeling stress?
  9. The discussion as a whole is well written, but also needs to be fleshed out.

Minor issues:

Please check the spelling of the p-values (should be the same p - italics everywhere)

Line 37-44 - add literature references (only one reference on glucose metabolism).

Line 71-80 - the logic of presentation and the need for this paragraph are not entirely clear

Line 244 - No literature reference

Little things that need to be fixed.

I would like to see a more visual representation of the data (not just bar graphs, but results confirming differences in gene expression) and confirmation of the data by several methods.

Author Response

Response to Reviewer 3 Comments

In the paper by Mariola Herbet et al., titled “Alteration in the expression of genes involved in brain energy metabolism as a process of adaptation to stressful conditions” the authors describe changes in brain energy metabolism as a process of adaptation to stressful conditions and may constitute the basis for the development of new therapeutic solutions for depression associated with chronic stress.

The article is written on an intriguing topic of modern neuroscience and has a scientific interest.

However, several issues need to be addressed.

Major issues:

Point 1: According to the authors, studying the expression of genes involved in energy brain metabolism, it can be concluded that they are involved in adaptation to stressful conditions. The authors are welcome to present more data on changes in glucose/energy metabolism (expression of enzymes involved in the regulation of energy metabolism, lactate, mitochondria, etc.).

Response 1: We would like to thank the Reviewer for comments and suggestion, which helped us improve the quality of our manuscript. After considering the comments of the Reviewers, we decided to re-edit the title of the work and the introduction. Our research focused on glucose-dependent brain energy metabolism. However, this was not emphasized in the original version of the manuscript. Therefore, in this new version, we focused on cerebral glucose metabolism; we also mentioned alternative energy sources for the brain.

Point 2: The introduction is very long, but does not specify what the focus will be on. It makes sense to describe in more detail the choice of genes for research and describe what mechanism they plan to study.

Response 2: As suggested by the Reviewer, the introduction has been redrafted. We also added information about the selection of genes for research and the mechanisms studied.

Point 3: In the introduction, it makes sense to supplement information about the current state of the problem of energy metabolism in the brain.

Response 3: As mentioned above, the introduction has been redrafted.

Point 4: Line 81-93 – clearly highlight the main idea and focus of the study

Response 4: This excerpt has been reworded and shortened.

Point 5: It is not entirely clear the choice of the stress model. Why did you choose a corticosteron administration, and not other models (early-life stress etc). Reference to article [17] did not provide a description of the model. Please clarify.

Response 5: The repeated administration of corticosterone leads to excessive activity of the hypothalamic-pituitary-adrenal axis and causes behavioural, as well as, neurochemical changes in the brain of animals adequate to the symptoms and neurochemical changes co - existing with clinical depression in humans. We chose this animal paradigm for our experiments because the repeated corticosterone injection paradigm provides a useful and reliable mouse model within which to further study the role of stress in depressive illness, as well as screen for antidepressants or preventive drugs. We plan to conduct such research in the future.

There was a citation error in the manuscript - this has been corrected. We are sorry and thank you very much for this remark.

Zhao Y, Ma R, Shen J, Su H, Xing D, Du L. A mouse model of depression induced by repeated corticosterone injections. Eur J Pharmacol. 2008 Feb 26;581(1-2):113-20. doi: 10.1016/j.ejphar.2007.12.005. Epub 2007 Dec 14. PMID: 18184609.

Point 6: 2 Experimental project: I recommend adding a design scheme. The description of the stress model is best placed in the methods section, rather than in the introduction.

Response 6: As suggested by the Reviewer, the design scheme has been prepared. The detailed description of the model used has been removed from the introduction.

Point 7: Why gene expression was assessed only in the cortex and not in the hypothalamus, pituitary gland, amygdala (fear) or other regions?

Response 7: The role of specific brain regions in the clinical pathophysiology of depression is poorly understood. However, the prefrontal cortex appears to be directly involved in clinical depression. The prefrontal cortex has emerged as one of the region’s most consistently impaired in major depressive disorder. Functional imaging studies showed hypo metabolism of the prefrontal lobe in primary and secondary depression. Research suggests that prefrontal cortex dysfunction has the potential to trigger many of the symptoms seen in clinical depression.

Mark S. George M.D.,Terence A. Ketter M.D.,Dr. Robert M. Post M.D. Prefrontal cortex dysfunction in clinical depression. doi/10.1002/depr.3050020202

Pizzagalli DA, Roberts AC. Prefrontal cortex and depression. Neuropsychopharmacology. 2022 Jan;47(1):225-246. doi: 10.1038/s41386-021-01101-7. Epub 2021 Aug 2. Erratum in: Neuropsychopharmacology. 2021 Aug 19;: PMID: 34341498; PMCID: PMC8617037.

Kennedy SH, Evans KR, Krüger S, Mayberg HS, Meyer JH, McCann S, Arifuzzman AI, Houle S, Vaccarino FJ. Changes in regional brain glucose metabolism measured with positron emission tomography after paroxetine treatment of major depression. Am J Psychiatry. 2001 Jun;158(6):899-905. doi: 10.1176/appi.ajp.158.6.899. PMID: 11384897.

Point 8: Did you measure basal levels of corticosterone and glucose and after modeling stress?

Response 8: The experiment was carried out on mice from certified farms and the animals were randomly assigned to groups. We made no measurements at the beginning of the experiment; we assumed that all animals are healthy and without any deviation from the norms of basic parameters (which is guaranteed by certain requirements to be met by a laboratory animal breeding centre; Experimental Medicine Center with the Certificate of Good Laboratory Practice; Lublin; Poland). Since we used an animal model in which mice were administered corticosterone for 21 days, we did not plan to determine its level after the end of the experiment. The obtained results could not be clearly interpreted. It would be unclear what the corticosterone pool is due to exogenous administration and what is the endogenous hormone pool. Behavioural tests and body weight measurements performed after its completion were the confirmation of the assumed changes in the adopted model (described in the scientific literature). We have adopted the hypothesis that the use of a proven animal model, confirmed by behavioural research, leads to changes in the level of hormones in the body, which has been the subject of many studies. The aim of our study was to evaluate the expression of selected genes involved in glucose metabolism in the brain during chronic stress. Such research is limited and the scientific literature lacks detailed data on this subject. We agree with the Reviewer that the determination of the levels of hormones and/or glucose in animals would be an additional, valuable confirmation of our assumptions, however, they were not made. It is currently not possible to perform additional determinations; we will consider this consideration in the planning of future experiments. However, in our opinion, the results obtained in the study are valuable, provide new information and should be published.

Point 9: The discussion as a whole is well written, but also needs to be fleshed out.

Response 9: The discussion chapter has been improved.

Minor issues:

Point 10: Please check the spelling of the p-values (should be the same p - italics everywhere)

Response 10: It has been standardized.

Point 11: Line 37-44 - add literature references (only one reference on glucose metabolism).

Response 11: Relevant references have been added.

Point 12: Line 71-80 - the logic of presentation and the need for this paragraph are not entirely clear

Response 12: This fragment has been removed from the introduction.

Point 13: Line 244 - No literature reference

Response 13: Relevant references have been added.

Point 14: I would like to see a more visual representation of the data (not just bar graphs, but results confirming differences in gene expression) and confirmation of the data by several methods.

Response 14: In our study, we intended to find out if there was a change in gene expression in the prefrontal cortex of mice under conditions of chronic stress. We consider the expression of selected genes to be some kind of marker of what happens in the brain under experimental conditions; this has been confirmed by our research. In our humble opinion, the visual presentation of the obtained results was done in an adequate, clear and readable way, similar to other scientific articles on this issues as well as our recently published studies.  We agree with the Reviewer's suggestion that the presented data could be confirmed by other methods, but it was not planned and performed in this study and it is currently not possible to perform additional determinations. We will take this valuable suggestion into account when planning our future studies. Often, the obtained results of a research urge to continue investigation of a certain issue in further studies. Therefore, in order to avoid collecting the materials for publication for years, the current achievements should be published and further researches will be presented in a next manuscript.

Round 2

Reviewer 1 Report

Authors have satisfactorily addressed most of the comments . Therefore, I would suggest to consider this manuscript for publication.

Reviewer 2 Report

The authors have improved the manuscript taking into account the recommendations and therefore I consider that the study  meets the minimum requirements necessary for its publication.

Reviewer 3 Report

The authors have worked on the article, that has significantly improved it.

This manuscript is a resubmission of an earlier submission. The following is a list of the peer review reports and author responses from that submission.